# Optimal Option of n-Level Polybinary Transformation in Faster than Nyquist System According to the Time-Packing Factor

Peng Sun, Wenbo Zhang, Dongwei Pan and Xiaoguang Zhang *

State Key Laboratory of Information Photonics and Optical Communications, Beijing University of Posts and Telecommunications, Beijing 100876, China
* Correspondence: xgzhang@bupt.edu.cn

**Abstract:** According to an in-depth analysis of the relationship among n-level polybinary transformation, the time-packing factor and the performance of the decoding algorithm, we find that the appropriate n-level polybinary transformation can improve the performance of the decoding algorithm within a certain range of the time-packing factor in the Faster than Nyquist (FTN) system. In this paper, we explain the reason that this phenomenon occurs. Based on the above analysis, we propose a modified blind phase search (BPS) algorithm to compensate for phase noise (PN) in the FTN system with an extremely small time-packing factor. As a result, the modified-BPS algorithm can cope with the PN with the linewidth $\times$ symbol rate at $1.07 \times 10^{-5}$, $1.79 \times 10^{-5}$, $2.86 \times 10^{-5}$ and $3.57 \times 10^{-5}$ under a time-packing factor of 0.55, 0.50 and 0.45, respectively. At the same time, the spectrum efficiency (SE) is improved to 3.27 bit/s/Hz, 4 bit/s/Hz and 4.88 bit/s/Hz.

**Keywords:** polybinary transformation; time-packing factor; decoding algorithm; modified-BPS algorithm

## 1. Introduction

The spectrum efficiency (SE) is one of the most significant parameters in optical communication systems; hence, the question of how to improve the SE is crucial [1–6]. Several methods can be adopted to increase the SE, in which the Faster than Nyquist (FTN) technique is one of the most favorable methods. In the FTN system, the ratio between the signal's bandwidth and the baud rate is called the time-packing factor, indicating the amount of increase in SE that can be obtained. With a given baud rate, the bandwidth reduction in the occupied signal means a higher SE, while, at the same time, it induces inter-symbol interference (ISI). Generally, ISI needs to be mitigated by decoding algorithms, such as the maximum-likelihood-sequence estimation (MLSE) algorithm [7] and Bahl–Cocke–Jelinek–Raviv (BCJR) algorithm [8].

Since Dr. Mazo in the Bell Lab proposed the concept of the FTN technique in 1975 [9], researchers worldwide have performed a great deal of work [10–14]. In the aforementioned works relating to FTN, the values of the bandwidth compression ratio or the time-packing factor adopted are 0.89 [10], 0.75 [11], 0.87 [12], 0.90 [13] and 0.7 [14], respectively. All used values are not less than 0.7. A smaller time-packing factor means a higher SE, with the sacrifice of the more serious ISI, hence causing the performance of the used decoding algorithms to degrade or even causing their failure.

The problem whereby we cannot realize a smaller time-packing factor in the FTN system reduces the benefits of the deployment of the FTN system. Therefore, the question of how to ensure that decoding algorithms are still effective under a smaller time-packing factor is of great significance to the FTN system. In recent years, research works on polybinary transformation have become more attractive [15,16]. An adaptive detection technique was proposed, which utilizes polybinary transformation converting the QPSK signal to n-level polybinary signals, in order to realize the signal's detection for an FTN system with a small time-packing factor. The numerical simulation and experiment proved

that the technique could cope with a 0.53 and 0.71 time-packing factor for the FTN-QPSK system [15]. The concept of non-orthogonal WDM was proposed, which, respectively, realized the detection of a 112GBaud 8QAM signal and a 83GBaud 16QAM signal in 100 GHz and 75 GHz frequency grids by using polybinary transformation, and the time-packing factors, respectively, were 0.89 and 0.93 [16]. Therefore, polybinary transmission can enhance the performance of decoding algorithms for the FTN system with a small time-packing factor. However, the research works in the literature do not clearly explain the relationship among the polybinary transformation, time-packing factor and performance of decoding algorithms.

In this paper, we analyze the relationship among the polybinary transformation, time-packing factor and performance of the decoding algorithms, in order to determine the influence of an extremely small time-packing factor on the performance of the decoding algorithm. Based on the above analysis, we find that the method of performing an n-level polybinary transformation combined with a designed MLSE results in better performance than that without doing so. However, it is worth noting that when the time-packing factor changes from 1.00 to 0.45, the best working range of the method will be quite different. Based on the above analysis, we propose a modified-BPS algorithm to compensate for PN in the FTN system under a smaller time-packing factor. Utilizing the modified-BPS algorithm, we find that this method can equalize the PN with the linewidth symbol rate at $1.07 \times 10^{-5}$, $1.79 \times 10^{-5}$, $2.86 \times 10^{-5}$ and $3.57 \times 10^{-5}$, in which the time-packing factor equals 0.55, 0.50 and 0.45, respectively.

## 2. The Principles of Polybinary Transformation

An n-level polybinary transformation can be completed by a series of delay operations and modulo-two sum operations. The operation of n-level polybinary transformation can be expressed as two steps as follows.

The first step is to transform the original input binary message sequence $\{a_m\}$ into another binary sequence $\{d_m\}$. The present binary sequence of $\{d_m\}$ is generated by forming the modulo-two sum of the present input sequence of $\{a_m\}$ and the preceding n-2 of $\{d_m\}$. The above-stated mathematical calculation can be expressed as an equation:

$$d_m = a_m \oplus d_{m-1} \oplus d_{m-2} \oplus \cdots \oplus d_{m-n+2} \tag{1}$$

where $\oplus$ represents the exclusive-or logic operation (modulo-two addition).

The second step is to generate an n-level polybinary sequence $\{p_m\}$, and the ith element of this ploybinary sequence $p_i$ is proportional to the algebraic sum of the successive n-2 values of the binary sequence $\{d_m\}$. This process is expressed by

$$p_i = E \sum_{k=0}^{n-2} d_{m-k} \tag{2}$$

The structure for these operations is illustrated in Figure 1, where $\{a\}$ is the input binary message sequence, $\{d\}$ is the transformational binary sequence, $E$ is an arbitrary constant, $\{p\}$ is the n-level polybinary sequence at the detector and $T$ is the period of the input binary message sequence.

An n-level polybinary transformation has its name due to the aforementioned n-2 symbol period delays, in which 2-level and 3-level polybinary transformation have their conventional names of duobinary transformation and tribinary transformation, respectively. Note that 1-level polybinary transformation does not involve any operation on the original signal, and it has no importance.

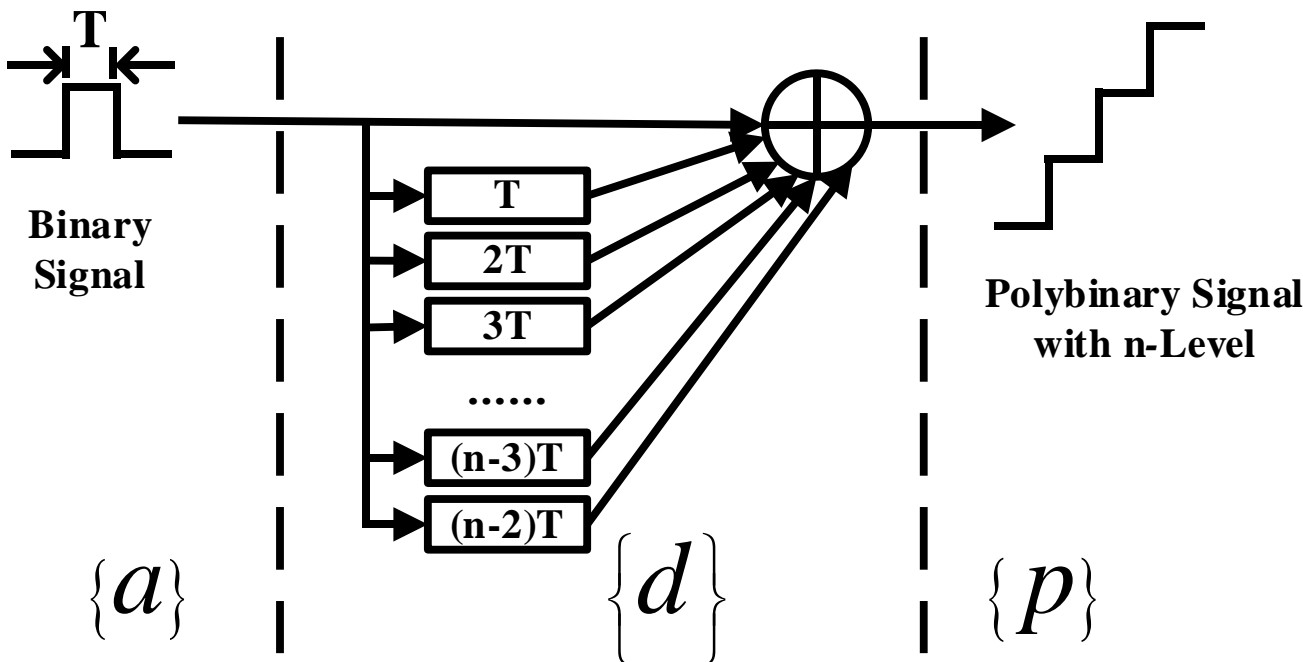

**Figure 1.** Structure of n-level polybinary signal by delay operations and modulo-two sum operations.

Figures 2a–c and 3a–c are the probability density figures (PDFs) of QPSK and 16QAM constellations when performing 1-level polybinary transformation, duobinary transformation and tribinary transformation, respectively. The colors from red to dark blue represent the level of probability density of the constellation points; the red color denotes a high level and the blue color denotes a low level. Besides the aforementioned phenomena, the number of constellation points changes after the polybinary transformation. Figure 2 shows that the number of constellation points changes from 4 points to 9 points and 16 points when the original QPSK signal undergoes duobinary transformation and tribinary transformation, respectively. Figure 3 shows that the number of constellation points changes from 16 points to 49 points and 100 points for the original 16QAM signal after duobinary transformation and tribinary transformation, respectively.

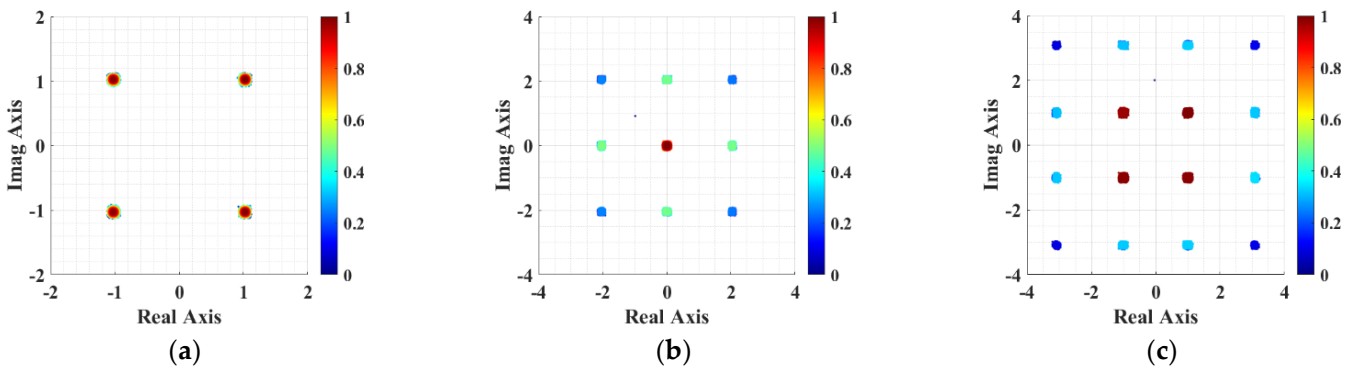

**Figure 2.** The PDF of n-level polybinary transformation for QPSK signal: (**a**) 1-level polybinary transformation of original signal; (**b**) duobinary transformation; (**c**) tribinary transformation.

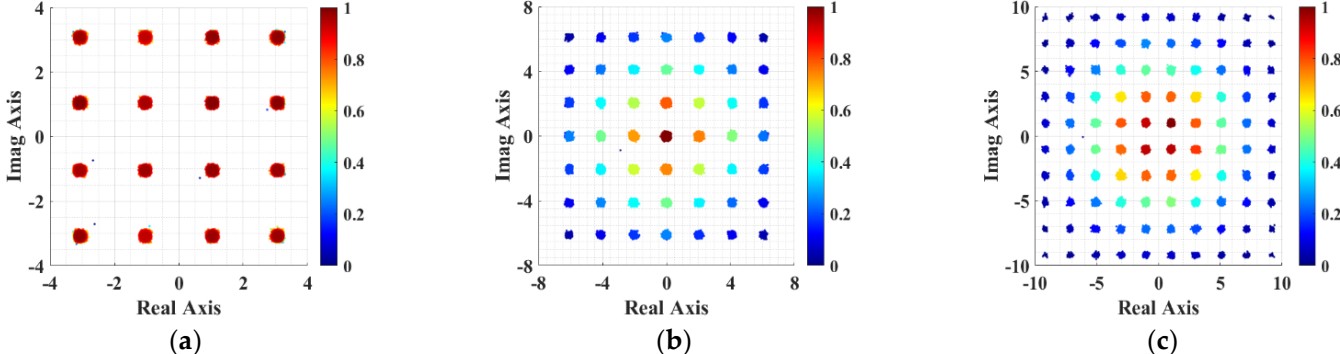

**Figure 3.** The PDF of n-level polybinary transformation for 16QAM signal: (**a**) 1-level polybinary transformation of original signal; (**b**) duobinary transformation; (**c**) tribinary transformation.

Note that the above processes for the bit message sequence and the symbol message sequence are similar. Therefore, the polybinary transformation can be achieved either at the transmitter side or receiver side in a practical system. Compared with the polybinary transformation at the transmitter side, the polybinary transformation at the receiver side can be combined with DSP algorithms to efficiently equalize the impairments. Therefore, we will look into the advantages and benefits of polybinary transformation at the receiver side in this paper.

### 3. The Relationship among Time-Packing Factor, n-Level Polybinary Transformation and Performance of Decoding Algorithms

Using the FTN technique in a system means that higher SE is obtained with a smaller time-packing factor. However, an extremely small time-packing factor induces more serious ISI in the received signal sequence, hence increasing the burden of the decoding algorithm and even making the algorithm unworkable. Obtained through numerical simulation, Figure 4a shows the PDF diagram of the original QPSK signal when the time-packing factor equals 0.50. In this case, the SE doubles but the ISI is severely induced, with the result that we cannot clearly distinguish the constellation points. Therefore, we utilize the n-level polybinary transformation for the original QPSK signal with the time-packing factor of 0.50. Figure 4b,c show the PDF diagrams after duobinary transformation and tribinary transformation, respectively.

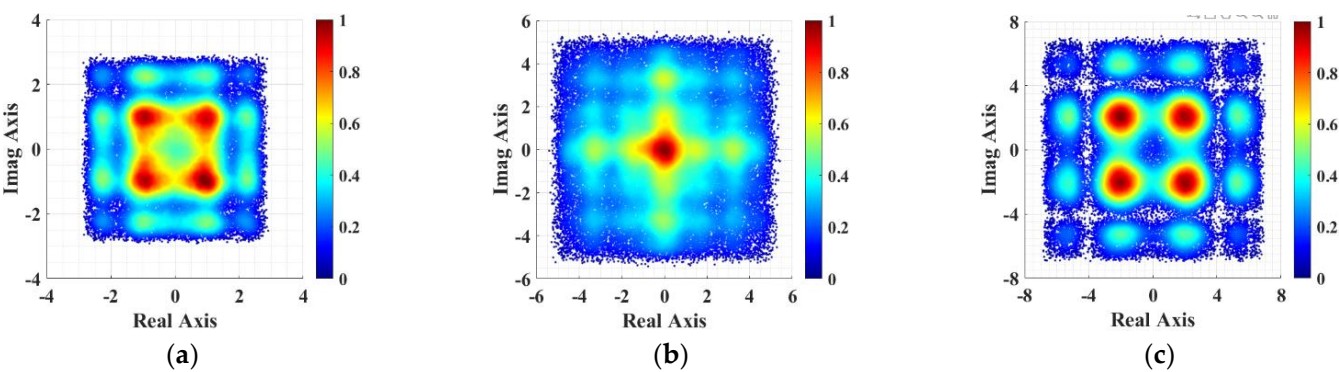

**Figure 4.** The PDF of n-level polybinary transformation for QPSK signal under the time-packing factor of 0.5: (**a**) original QPSK signal; (**b**) after duobinary transformation; (**c**) after tribinary transformation.

Compared with Figure 4a,b, we find that after duobinary transformation, the serious overlap of the constellation points makes them difficult to distinguish. In this case, the decoding algorithm cannot work effectively. However, if we utilize tribinary transformation, the situation is improved. As shown in Figure 4c, the overlapping phenomenon seems greatly reduced. After using tribinary transformation instead of duobinary transformation,

as in Figure 4c, we can see that the 16 clusters are clearly distinguishable and the subsequent decoding algorithms will complete the decoding work. Based on the above discussions, we can find that an appropriate n-level polybinary transformation can somewhat solve the ISI problems in the FTN system. However, when using the incorrect transformation, we cannot achieve the elimination of ISI in the FTN signal, instead aggravating the ISI problem. Therefore, we need to explore how to use the appropriate n-level polybinary transformation and achieve the best performance of the decoding algorithm.

We build a 28GBaud PDM-FTN-QPSK system as a simulation platform, as shown in Figure 5, to test the influence of n-level polybinary transformation on the performance of MLSE under the different time-packing factors. The roll-off factor is 0.15 for the model of pulse shaping. We assume that there exist no frequency offset (FO) or PN for the transmitter laser and local laser. We also assume that the channel impairments, such as chromatic dispersion (CD), polarization mode dispersion (PMD) and rotation of state of polarization (RSOP), have been equalized by corresponding DSP algorithms. (We have conducted work to jointly equalize PMD and RSOP in the FTN system under a small time-packing factor [17]. In this paper, we only focus our attention on coping with the problem of ISI induced in the FTN system.) Therefore, the remaining impairments that we considered only included ISI and ASE noise in the FTN system.

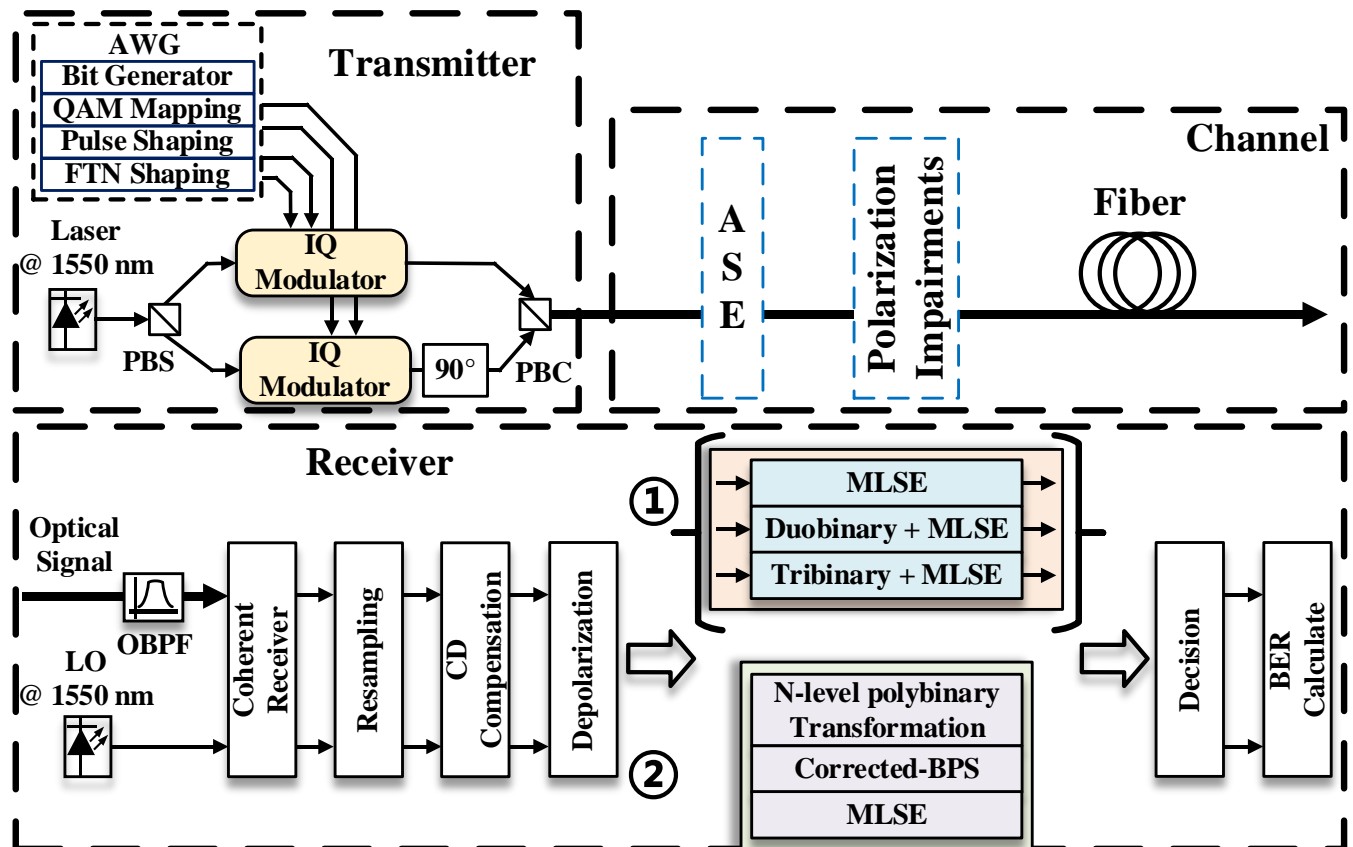

**Figure 5.** Simulation platform for 28GBaud PDM-FTN-QPSK system.

As we have discussed, the appropriate n-level polybinary transformation is a prerequisite for the MLSE algorithm to solve the signal's ISI. Next, we will present the comparisons. We let the signal pass through three combinations of n-level polybinary transformation and subsequent MLSE. In other words, we allow the signal to pass through module ① in Figure 5. The first combination is 1-level polybinary transformation + MLSE. This means that only MLSE is used to implement ISI elimination. The second combination is 2-level polybinary transformation + MLSE (the equivalent name is duobinary transformation + MLSE). The third combination is 3-level polybinary transformation + MLSE (the equivalent name is tribinary

transformation + MLSE). The comparison of the aforementioned three combination algorithms' performance for the elimination of ISI is shown in Figure 6. Compared with the other two combinations, we can find that MLSE (0.95~1.00 time-packing factor), duobinary transformation + MLSE (0.55~0.90 time-packing factor) and tribinary transformation + MLSE (0.45~0.50 time-packing factor) have better performance, as shown in Figure 6a–c. Why does the phenomenon in which the combination of n-level polybinary transformation and MLSE leads to different decoding performance under different time-packing factors occur? We will explain this phenomenon as follows to ensure that we choose the appropriate n-level polybinary transformation for eliminating ISI under a certain time-packing factor.

In order to conduct an in-depth analysis, we attempt to convert the two-dimensional constellation PDF along with the real and imagined axes in Figures 2 and 4 into a one-dimensional constellation projection PDF only along with the real axis, as shown in Figure 7. We take four cases in which the time-packing factor equals 1.00, 0.95, 0.65 and 0.45 as examples, respectively. Figure 7a–d show the abovementioned constellation projection PDF graphs corresponding to the time-packing factors of 1.00, 0.95, 0.65 and 0.45, respectively. Note that, in Figure 7a–d, the received signals are both exercised with 1-level polybinary transformation (equivalent to the original signal), duobinary transformation and tribinary transformation.

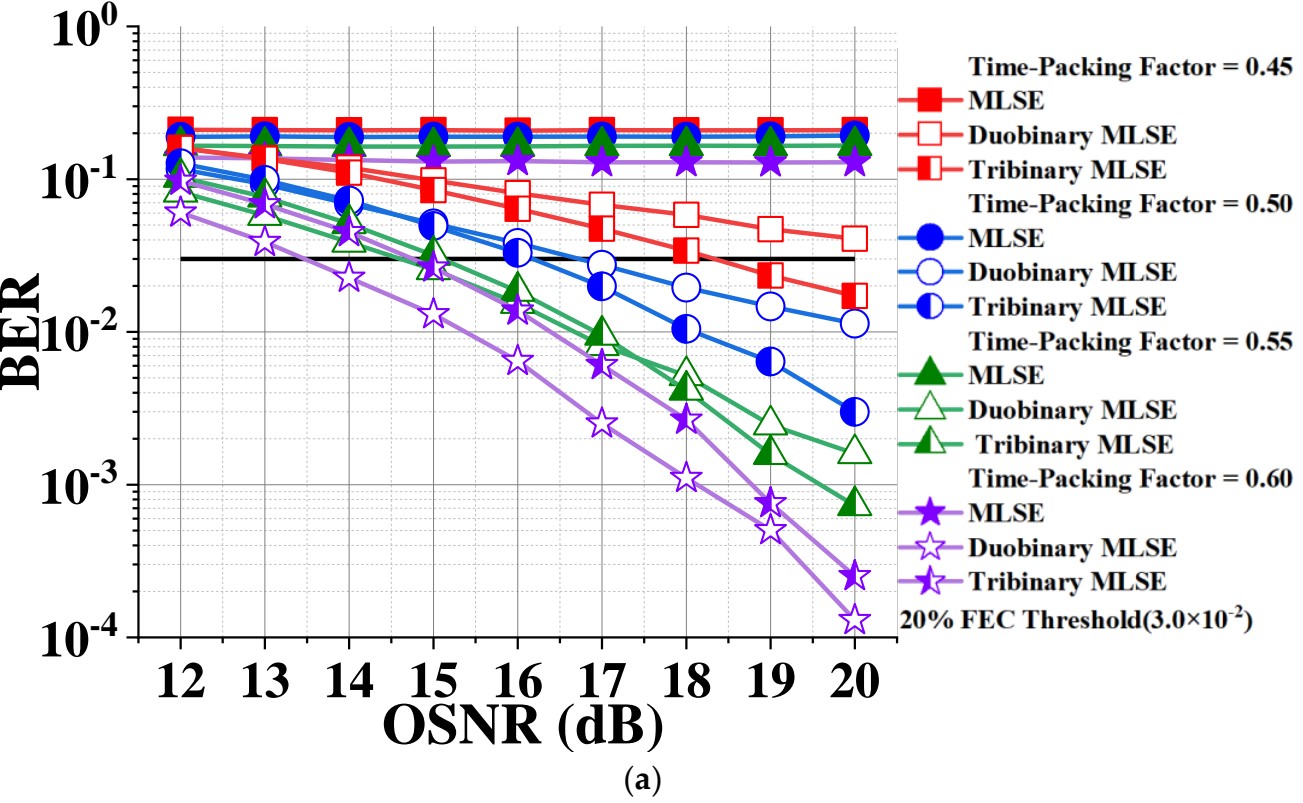

**Figure 6.** *Cont.*

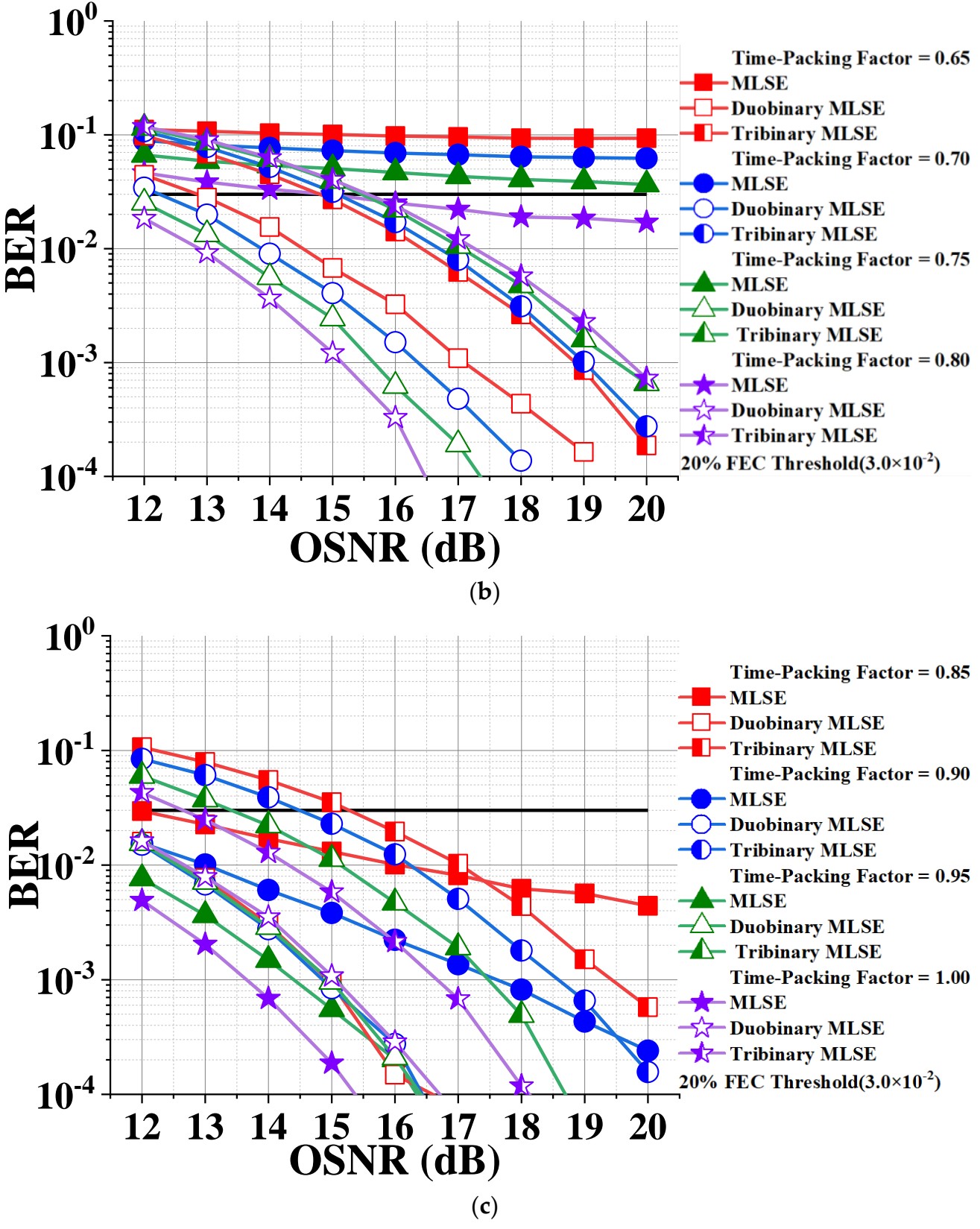

**Figure 6.** The performance of OSNR versus BER for MLSE, duobinary MLSE and tribinary MLSE under different time-packing factors: (**a**) the time-packing factor changes from 0.45 to 0.60; (**b**) the time-packing factor changes from 0.65 to 0.80; (**c**) the time-packing factor changes from 0.85 to 1.00.

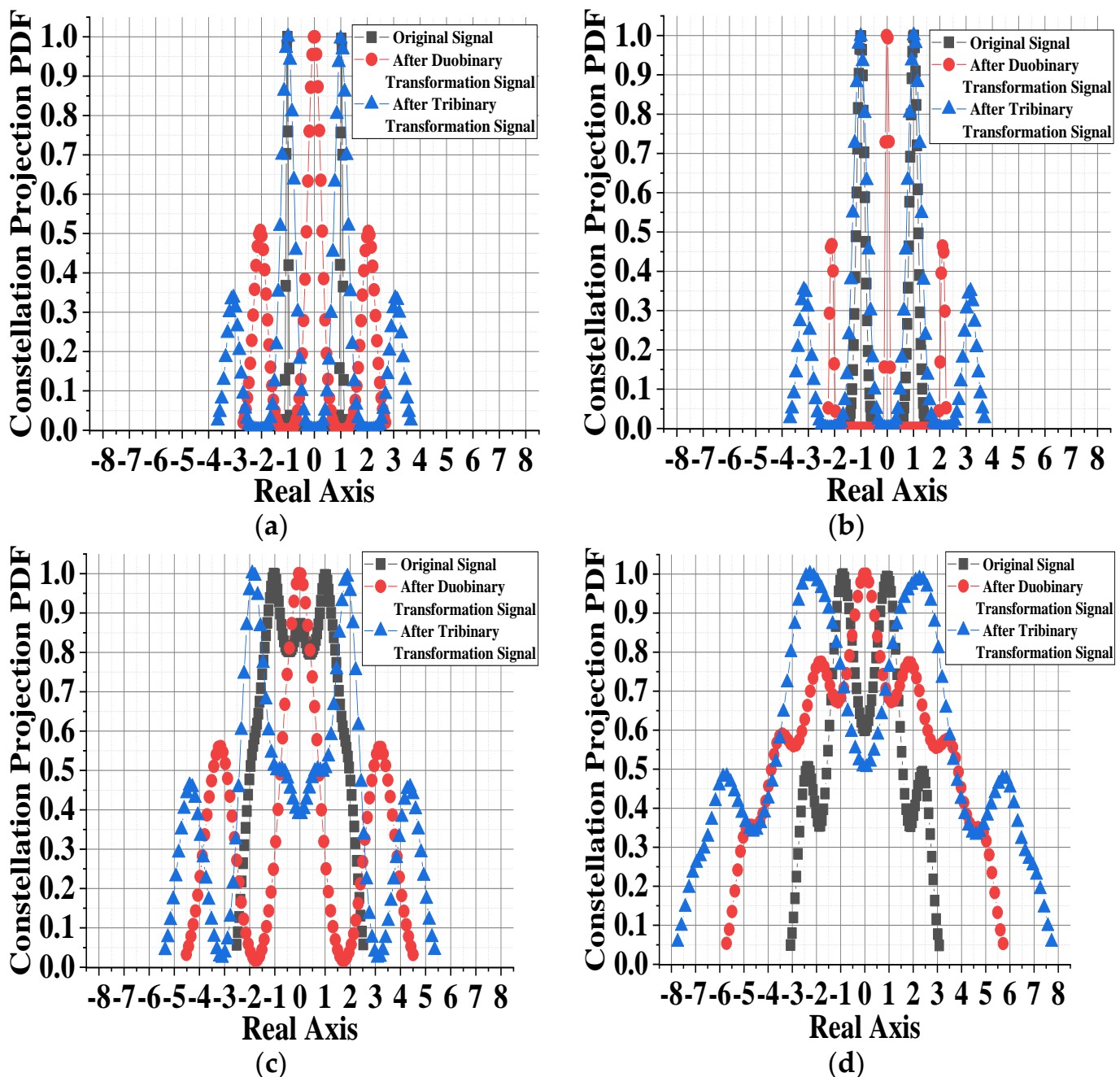

**Figure 7.** The normalized frequent and continuous versus real values of constellation points for original signal, after duobinary transformation and after tribinary transformation (**a**) when the time-packing factor equals 1.00; (**b**) when the time-packing factor equals 0.95; (**c**) when the time-packing factor equals 0.60; (**d**) when the time-packing factor equals 0.45.

The signal has no distortion effect induced by ISI with the time-packing factor of 1.00. Therefore, to analyze the system conveniently, we set the case with the time-packing factor of 1.00 as the standard benchmark for the following comparison, as shown in Figure 7a. The original signal has a 4 QAM constellation projection PDF curve, the duobinary transformed signal has a 9 QAM constellation projection PDF curve, and the tribinary transformed signal has a 16 QAM constellation projection PDF curve, as shown in Figure 7a. In Figure 7a, the three curves are the standard benchmark. For the smaller time-packing factors, the comparison results and analysis are shown as follows.

(1)     When the time-packing factor equals 0.95, in which little distortion is induced by ISI, we observe that all the curves in Figure 7a,b are similar. This means that with the time-packing factor of 0.95, we achieve almost the same performance with the combinations of 1-level polybinary transformation + MLSE (or MLSE), duobinary transformation + MLSE and tribinary transformation + MLSE, indicating that they can all eliminate ISI in the FTN system with the 0.95 time-packing factor. However, Figure 6c shows that the MLSE has better decoding performance with the 0.95 time-packing factor. Therefore, the MLSE is more suitable for the 0.95 time-packing factor.

(2)     When the time-packing factor equals 0.65, which is the case with medium ISI distortion, we observe that the original signal curve has a central peak (which is approximately 0.87), as shown in Figure 7c, which does not appear when the time-packing factor equals 1.00, indicating apparent distortion induced by ISI. Moreover, the tribinary transformed signal curve generates two maximum peaks at ±2, while the peaks at ±1 are severely decreased. Thus, we speculate that using the combinations of 1-level polybinary transformation + MLSE (or MLSE) and tribinary transformation + MLSE cannot achieve good decoding performance. However, we see that the duobinary transformed signal curve has the same locations of all peaks, as is the case of factor 1.00, as shown in Figure 7a,c. Hence, we suggest that it is appropriate to adopt the combination algorithm of duobinary transformation + MLSE, in order to achieve good decoding performance. All the abovementioned speculations are verified by the results in Figure 6b. Therefore, the duobinary transformation + MLSE is more suitable for the case of a 0.65 time-packing factor.

(3)     When the time-packing factor equals 0.45, which means severe ISI distortion, we observe in Figure 7d that all the transformed signal curves are largely deformed. We see that the number of curve peaks has changed for the 1-level polybinary transformed and duobinary transformed signals, compared with the case of a factor of 1.00. Moreover, the curve corresponding to the tribinary transformed signal also has four unchanged peaks, although the curve is apparently deformed. Therefore, we speculate that we can achieve good decoding performance when using the combination algorithm of tribinary transformation + MLSE, and it is not a wise choice to use 1-level polybinary transformation + MLSE or duobinary transformation + MLSE. These statements are also verified by the results in Figure 6a. Therefore, the tribinary transformation + MLSE is more suitable for the case of a 0.45 time-packing factor.

For other time-packing factors, we find similar relations to those mentioned above. Based on these relations, we can find the appropriate combinations of n-level polybinary transformation and MLSE in order to achieve the best decoding performance with certain time-packing factors. In this paper, we hope to enhance the SE further by adopting as small a time-packing factor as possible in the FTN system. Therefore, in the next section, we will focus on the case of 0.55~0.45 time-packing factors. The SE can be enhanced up to approximately 3.27 bit/s/Hz~4.89 bit/s/Hz under this time-packing factor range.

## 4. The Performance of Phase Noise Equalizing Algorithm in FTN System under Extremely Small Time-Packing Factor

All the discussions in the previous section are made under the assumption of no FO and no PN, which is not the case in a real optical fiber communication system. In this section, the received signal will pass through module ② in Figure 5. The PN induced by the linewidth of the laser both at the transmitter side and receiver side will influence the final quality of the decoding signal, which needs to be equalized by related algorithms. Generally, the most effective algorithm is the BPS algorithm, which can equalize PN for QPSK signals and other QAM signals, such as 16QAM, 32QAM, etc. BPS can achieve optimal performance based on the exact knowledge of the constellation point locations in the QAM signal. Because a smaller time-packing factor induces severe ISI and causes the serious overlap of constellation points, BPS will face the challenge of performance degradation in the FTN system. The lesser the overlap, the better performance of BPS can

be achieved in the FTN system. Moreover, the BPS is placed in the front of the decoding algorithm. In other words, the effectiveness of BPS will affect the performance of subsequent decoding algorithms. We need to explore an effective method to make the BPS algorithm workable again in the FTN system.

We take the 0.50 time-packing factor as an example. We find that the constellation points are not distinguishable in Figure 4a. Therefore, when we choose the original signal as the input signal with the 0.50 time-packing factor, the BPS will work ineffectively. According to the aforementioned analysis, we know that the tribinary transformation can greatly reduce the influence of ISI on the original QPSK signal when the time-packing factor equals 0.50. As shown in Figure 4c, based on the tribinary transformation, the constellation points become much more distinguishable, with the result that the BPS works again in the FTN system. To allow the BPS algorithm to work effectively in the FTN system, we need to make some modifications of the BPS algorithm according to the following two aspects: ① the number of constellation points should be changed from 4 to 16 (4-QAM to 16-QAM); ② the ideal locations of the constellation points should also be modified, and the new, ideal locations of constellation points should be expressed as in Equation (3), from that of 4-QAM to 16-QAM. For the sake of clarity, we call this type of BPS algorithm the modified-BPS algorithm. With the above modifications, the modified-BPS algorithm can compensate for PN in the FTN system with a smaller time-packing factor, as small as 0.50. Then, the MLSE algorithm can effectively achieve sufficient decoding performance to eliminate ISI. Therefore, when we make good use of the combination of the appropriate n-level polybinary transformation, the modified-BPS algorithm and MLSE, the received FTN QPSK signal can be recovered with different time-packing factors. Moreover, for other FTN QAM signals, such as 16QAM, 32QAM and 64QAM, similar methods can be adopted.

$$\begin{cases} 1+1j \\ -1+1j \\ -1-1j \\ 1-1j \end{cases} \Rightarrow \begin{cases} 2.1+2.1j, & 2.1+5.4j, & 5.4+5.4j, & 5.4+2.1j \\ -2.1+2.1j, & -2.1+5.4j, & -5.4+5.4j, & -5.4+2.1j \\ -2.1-2.1j, & -2.1-5.4j, & -5.4-5.4j, & -5.4-2.1j \\ 2.1-2.1j, & 2.1-5.4j, & 5.4-5.4j, & 5.4-2.1j \end{cases} \tag{3}$$

To verify the performance of the modified-BPS algorithm for compensating for PN in the FTN system with a smaller time-packing factor, we perform simulations on our platform, as shown in Figure 5. Note that the received signal will pass through module ② in Figure 5, which means that we take the linewidth of the lasers into account. The linewidth of the laser at the transmitter is set to be 300 kHz, 500 kHz, 800 kHz and 1000 kHz, respectively. We also assume that other channel impairments besides ISI have been equalized by corresponding DSP algorithms. The time-packing factor changes from 0.45 to 0.55. In module ②, the sequence that the received signal passes through the algorithm is as follows: (1) firstly, we utilize the duobinary or tribinary transformation for the received signal to eliminate the overlap of the constellation points; (2) secondly, the signal passes through the modified-BPS to compensate for PN after duobinary or tribinary transformation; (3) thirdly, the signal passes through the MLSE algorithm to eliminate ISI after compensating for PN. Figure 8a–c show the BER as a function of OSNR, which demonstrates the effectiveness of the modified-BPS algorithm to equalize PN in relation to the linewidths of 300 kHz, 500 kHz, 800 kHz and 1000 kHz in the FTN system with the different time-packing factors. In this paper, the linewidth of laser $\Delta f$ is defined as the full width at half maximum (FHWM). Figure 9a,b show the required ONSR and OSNR penalty versus the product of the linewidth and symbol period ($\Delta f \cdot Ts$) for the modified-BPS algorithm in the FTN system with the different time-packing factors, respectively. We call the product of the linewidth and symbol period ($\Delta f \cdot Ts$) the linewidth symbol rate. The FEC threshold is $3.0 \times 10^{-2}$, corresponding to soft-decision 20% FEC.

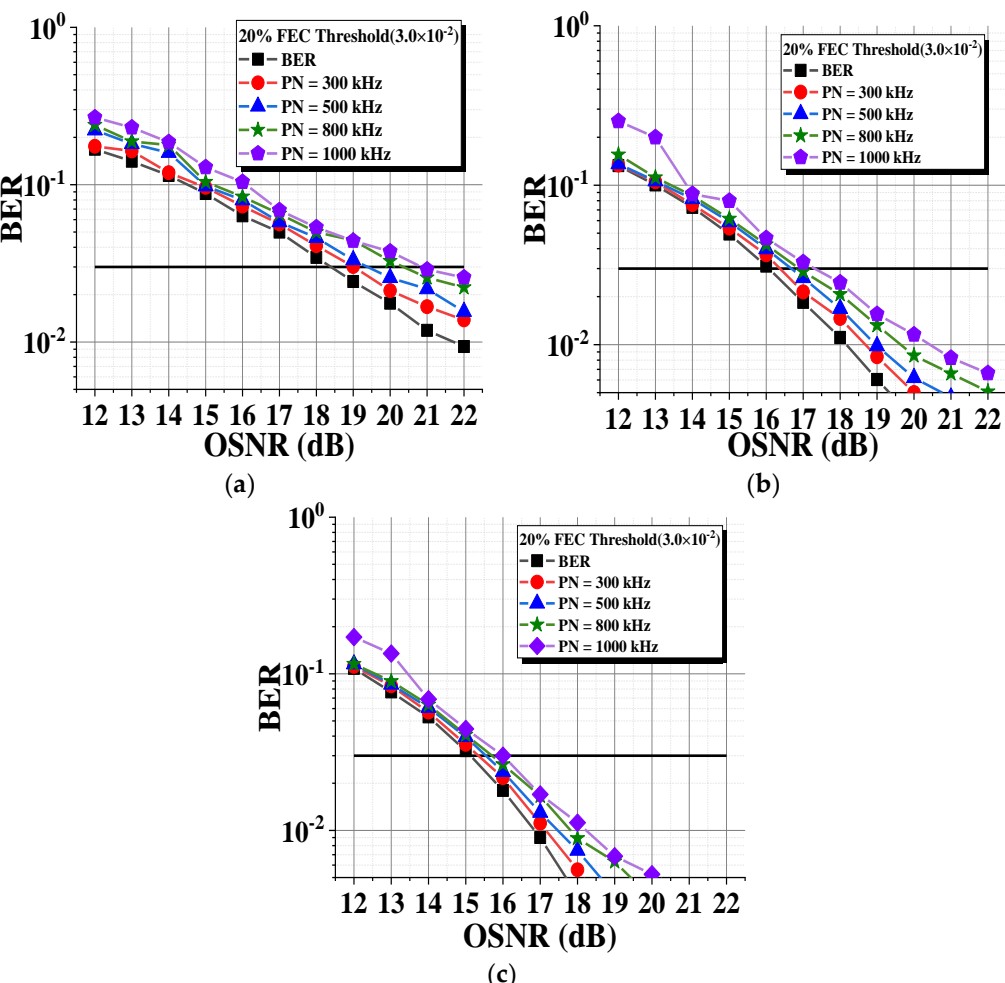

**Figure 8.** Under the different time-packing factors, the performance of OSNR versus BER is shown for the modified-BPS algorithm to equalize PN when the linewidth equals 300 kHz, 500 kHz, 800 kHz and 1000 kHz, respectively, and (**a**) the time-packing factor equals 0.45; (**b**) the time-packing factor equals 0.50; (**c**) the time-packing factor equals 0.55.

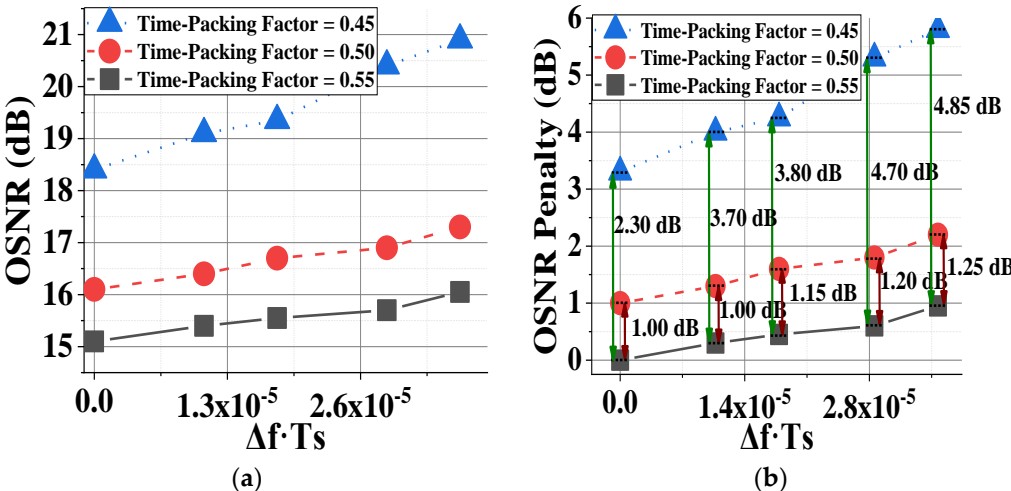

**Figure 9.** The performance of required OSNR and OSNR penalty versus linewidth symbol rate ($\Delta f \cdot Ts$) under the time-packing factor changes of 0.45~0.55: (**a**) required ONSR versus $\Delta f \cdot Ts$; (**b**) OSNR penalty versus $\Delta f \cdot Ts$.

We can see in Figure 8a–c that the modified-BPS algorithm can effectively compensate for 300 kHz, 500 kHz, 800 kHz and 1000 kHz linewidths with the time-packing factor of 0.45–0.55. Figure 9a shows the required OSNRs for the modified-BPS algorithm with the $\Delta f \cdot Ts$ parameter values of 0, $1.07 \times 10^{-5}$, $1.79 \times 10^{-5}$, $2.86 \times 10^{-5}$ and $3.57 \times 10^{-5}$ when the time-packing factor takes values within 0.45–0.55. To observe and compare the results conveniently, the required OSNRs for the modified-BPS algorithm with different time-packing factors and $\Delta f \cdot Ts$ are listed in Table 1. To measure the OSNR penalty, we set 15.10 dB as the standard benchmark that corresponds to the case of a time-packing factor of 0.55 and $\Delta f \cdot Ts$ of 0. Moreover, Figure 9b shows the OSNR penalties for the modified-BPS algorithm under the same conditions as Figure 9a, and the results are listed in Table 2.

**Table 1.** The required OSNRs for the modified-BPS algorithm under 0, $1.07 \times 10^{-5}$, $1.79 \times 10^{-5}$, $2.86 \times 10^{-5}$ and $3.57 \times 10^{-5}$ $\Delta f \cdot Ts$ with the time-packing factor of 0.45–0.55.

| $\Delta f \cdot Ts$    Time-Packing Factor<br>OSNRs/dB | 0.45 | 0.50 | 0.55 |
|---|---|---|---|
| 0 | 18.40 | 16.10 | 15.10 |
| $1.07 \times 10^{-5}$ | 19.10 | 16.40 | 15.40 |
| $1.79 \times 10^{-5}$ | 19.35 | 16.70 | 15.50 |
| $2.86 \times 10^{-5}$ | 20.40 | 16.90 | 15.70 |
| $3.57 \times 10^{-5}$ | 20.90 | 17.30 | 16.05 |

**Table 2.** The OSNR penalty for the modified-BPS algorithm under 0, $1.07 \times 10^{-5}$, $1.79 \times 10^{-5}$, $2.86 \times 10^{-5}$ and $3.57 \times 10^{-5}$ $\Delta f \cdot Ts$ with the time-packing factor of 0.45~0.55.

| $\Delta f \cdot Ts$    Time-Packing Factor<br>OSNR Penalty/dB | 0.45 | 0.50 | 0.55 |
|---|---|---|---|
| 0 | 3.30 | 1.00 | 0 |
| $1.07 \times 10^{-5}$ | 4.00 | 1.30 | 0.30 |
| $1.79 \times 10^{-5}$ | 4.25 | 1.60 | 0.45 |
| $2.86 \times 10^{-5}$ | 5.30 | 1.80 | 0.60 |
| $3.57 \times 10^{-5}$ | 5.80 | 2.20 | 0.95 |

In Table 1, we can see that the required OSNRs are 20.90 dB, 17.30 dB and 16.05 dB for $\Delta f \cdot Ts = 3.57 \times 10^{-5}$ under the time-packing factors equal to 0.45, 0.50 and 0.55, respectively. The corresponding OSNR penalties reach 5.80 dB, 2.20 dB and 0.95 dB, which are shown in Table 2, respectively. However, OSNR differences for 0 and $3.57 \times 10^{-5}$ $\Delta f \cdot Ts$ are 2.40 dB, 1.20 dB and 0.95 dB under the 0.45, 0.50 and 0.55 time-packing factors, respectively. Therefore, the performance of the modified-BPS algorithm is mainly influenced by the value of the time-packing factor rather than $\Delta f \cdot Ts$.

Hence, we can see that the modified-BPS algorithm can equalize effectively the PN in the FTN system with a smaller time-packing factor. Meanwhile, using the method of the combination of appropriate n-level polybinary transformation, modified-BPS and the MLSE algorithm can effectively recover the received FTN signals.

## 5. Conclusions

In this paper, we revealed the relationship among n-level polybinary transformation, the time-packing factor and the performance of the MLSE algorithm. Within a certain range of the time-packing factor, we find that the method of combining appropriate n-level polybinary transformation and MLSE can effectively eliminate serious ISI in the received FTN signals. With the addition of converting the two-dimensional constellation diagram

into the one-dimensional constellation projection PDF, we identify the reason that a different n-level polybinary transformation is required for different time-packing factors. For a time-packing factor within the large range of 1.00~0.45, we identify the three ranges of values within which the best n-level polybinary transformation should be adopted. Based on the above analysis, we propose a modified-BPS algorithm to compensate for PN in the FTN system with an extremely small time-packing factor. The proposed modified-BPS algorithm can effectively cope with the PN with the $\Delta f \cdot Ts$ at $1.07 \times 10^{-5}$, $1.79 \times 10^{-5}$, $2.86 \times 10^{-5}$ and $3.57 \times 10^{-5}$ under time-packing factors of 0.45, 0.50 and 0.55, respectively. In the case of the above conditions, the SE can be enhanced up to approximately 4.88 bit/s/Hz, 4 bit/s/Hz and 3.27 bit/s/Hz, respectively. It is noteworthy that, to verify the performance of our proposed scheme in the FTN system with an extremely small time-packing factor, we assume that FO, CD, PMD, RSOP and other impairments do not exist or have been equalized. However, in a large accumulated CD system, such as a long-haul transoceanic optical fiber communication system, we will encounter the problem of enhanced phase noise (EEPN), which is caused by the conflict between the long CD response time and short time along with the laser phase noise variation. Unlike CD or PMD impairments, there is no effective solution to mitigate this [18]. We will pay attention to the EEPN problem in our next work. In addition, in a real system, we must consider the hardware implementation efficiency, among which we should pay attention to the effective number of bits (ENOB) of the analog digital converter (ADC) or digital analog converter (DAC). In this paper, our proposed scheme is verified only at the level of software calculation with floating point numbers, to be capable of enhancing the performance of the decoding algorithm by using appropriate polybinary transformation, and the influence of ENOB is not considered. Therefore, we also need to explore the hardware implementation in the future.

**Author Contributions:** Conceptualization, P.S. and X.Z.; methodology, P.S. and W.Z.; software, P.S. and D.P.; data curation, P.S.; writing—original draft preparation, P.S; writing—review and editing, X.Z.; visualization, W.Z.; supervision, X.Z. All authors have read and agreed to the published version of the manuscript.

**Funding:** This research received was partly funded by the National Science Foundation of China (62071065).

**Conflicts of Interest:** The authors declare no conflict of interest.

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
