# Peer review of "Optimal Option of n-Level Polybinary Transformation in Faster than Nyquist System According to the Time-Packing Factor"

_applsci, doi:10.3390/app122111227_

Round 1

Reviewer 1 Report

In this paper, the authors provide a comprehensive overview on the fast-than-Nyquist (FTN) transmission.  The authors also demonstrate that an optimal polybinary transformation can improve the performance of FTN system.  The optimal performance is achieved when the probability distribution function (PDF) of signal after polybinary transformation is similar to the PDF of original signal.  This is the main conclusion from the paper.  In addition, the authors propose an improved algorithm of blind phase searching (BPS).  This improved algorithm takes into the consideration of optimal polybinary transformation.  The authors demonstrate the improved performance with the new algorithm.

In this paper, the authors provide good insight on improving FTN system with polybinary transformation.  The conclusions are drawn with good reasoning and solid data.  However, there are still many questions which require further answer.  There are many aspects which require clarifications and revision.  I’ve listed those below.  In my opinion, the paper can be accepted for publication with those questions being answered.

1. Optical signal can transmit over long distance and accumulate large amount of chromatic dispersion.  This is particularly true for QPSK signal, which is the main signal format under consideration for the paper.  The equalization of chromatic dispersion will enhance phase noise.  This is known as equalizer enhanced phase noise (EEPN).  Unlike CD or PMD, there is not an effective way to mitigate the impairment from EEPN [1].  So, this should be taken into the consideration.  How will EEPN influence the optimal choice of polybinary transformation?  [1] H. Sun et al., "800G DSP ASIC Design Using Probabilistic Shaping and Digital Sub-Carrier Multiplexing," in Journal of Lightwave Technology, vol. 38, no. 17, pp. 4744-4756, 1 Sept.1, 2020, doi: 10.1109/JLT.2020.2996188. 

2. Nyquist pulse shaping (NPS) can reduce the bandwidth of signal with sharp roll-off.  This will improve the spectral efficiency as well.  It is unclear whether FTN system described in this paper utilizes pulse shaping.  The authors should clarify.  Can pulse shaping be used in FTN system?  Will the conclusion be influenced if pulse shaping is used? 

3. The polybinary transformation will convert a QPSK signal into a high-order QAM signal.  For example, 9-QAM signal (duobinary) or 16-QAM signal (tribinary).  It is known that to achieve good performance on QAM signal, an ADC (or DAC) with higher effective number of bit (ENOB) is required.  Can the authors characterize the impact on ENOB when the polybinary transformation is used? 

4. Can the authors provide a definition of linewidth of laser?  Is it Lorentzian linewidth?  Or other definition? 

5. In Fig7, there are four sub plots.  However, only three plots are described in the figure caption below.  I believe that time packing factor of 7(a) is 1, 7(b) is 0.95, 7(c) is 0.65 and 7(d) is 0.45.  Please correct the figure caption. 

6. Are the constellation points in Eq. (3) obtained from the simulation result, like that shown in Fig. 4(c)?  Please clarify how one obtain the constellation points in the paper. 

7. In the abstract, there is no need to display so many digits as shown below.  Rounding the linewidth*symbolrate to two decimal digits like 1.07e-5, 1.79e-5, should be enough.

8. In the paper, there are many long sentences.  In general, readers can get confused by long sentences.  One suggestion is to break down a long sentence into a few short sentences.

Author Response

1. Optical signal can transmit over long distance and accumulate large amount of chromatic dispersion. This is particularly true for QPSK signal, which is the main signal format under consideration for the paper. The equalization of chromatic dispersion will enhance phase noise. This is known as equalizer enhanced phase noise (EEPN). Unlike CD or PMD, there is not an effective way to mitigate the impairment from EEPN [1]. So, this should be taken into the consideration. How will EEPN influence the optimal choice of polybinary transformation? [1] H. Sun et al., "800G DSP ASIC Design Using Probabilistic Shaping and Digital Sub-Carrier Multiplexing," in Journal of Lightwave Technology, vol. 38, no. 17, pp. 4744-4756, 1 Sept.1, 2020, doi: 10.1109/JLT.2020.2996188.

Thank you for your comment. EEPN is really a trouble problem in a large accumulated CD coherent detection system, and should be paid attention to. It is large topic that can be discussed in another paper like in Ref. [1] the reviewer has mentioned. But in this manuscript, the main focus is on the difference of equalization in the FTN QPSK and the ordinary QPSK system. Therefore we think that we might overlook EEPN problem in this manuscript. As mentioned in Ref. [1], the example taken is the system of CD 350000 ps/nm, which is equivalent to a standard single mode fiber of 350000 ps/nm ¸ 17 ps/nm/km = 20588 km. In our manuscript, the transmission distance is about several hundred kilometers, in which the EEPN is of not sufficient impact on the system. However, we do think this EEPN should be addressed in the manuscript in order for a reader to pay attention to. So we take the Ref.[1] as the one reference at the end of our manuscript, and mentioned the problem of EEPN in page 13 to remind the readers to pay attention to. Maybe, in the following work, we should explore the influence of EEPN on the FTN system. In Page 13, at the end of conclusion, we added a paragraph “The proposed modified-BPS algorithm can effectively cope with the PN with the Δf·Ts at 1.07E-5, 1.79E-5, 2.86E-5 and 3.57E-5 under time-packing factor of 0.55, 0.50 and 0.45, respectively. In the case of above conditions, the SE can be enhanced up to about 3.27 bit/s/Hz, 4 bit/s/Hz and 4.88 bit/s/Hz, respectively. It is noteworthy that, to verify the performance of our proposed scheme in FTN system with the extreme small time-packing factor, we assume FO, CD, PMD, RSOP and other impairments are not exist or have been equalized. However, in a large accumulated CD system such as long-haul transoceanic optical fiber communication system, we will face the trouble problem of enhanced phase noise (EEPN) which is caused by the conflict between the long CD response time and short time along with the laser phase noise variation. Unlike CD or PMD impairments, there is not an effective way to mitigate it [18]. We should pay attention to the EEPN trouble in our next works.[18] Han. Sun, Mehdi Torbatian, Mehdi Karimi, et al., "800G DSP ASIC Design Using Probabilistic Shaping and Digital Sub-Carrier Multiplexing," J. of Lightwave Technol. 38, 4744-4756 (2020).

2. Nyquist pulse shaping (NPS) can reduce the bandwidth of signal with sharp roll-off. This will improve the spectral efficiency as well. It is unclear whether FTN system described in this paper utilizes pulse shaping. The authors should clarify. Can pulse shaping be used in FTN system? Will the conclusion be influenced if pulse shaping is used?

Thank you for your comment. In our FTN system, we utilize the pulse shaping. The roll-off factor of pulse shaping will influence the number and value of constellation points after polybinary transformation. The system’s structure of simulation platform which shows in Figure 5 has been modified in paper. We also added a sentence in page 6 “The roll-off factor is 0.15 for the model of pulse shaping.

3. The polybinary transformation will convert a QPSK signal into a high-order QAM signal. For example, 9-QAM signal (duobinary) or 16-QAM signal (tribinary). It is known that to achieve good performance on QAM signal, an ADC (or DAC) with higher effective number of bit (ENOB) is required. Can the authors characterize the impact on ENOB when the polybinary transformation is used?

Thank you for your comment. As for equalization algorithms, we will evaluate them effective in terms of two levels, one is on the level of software, and on the level of hardware implementation. On the level of software, we assume that ADC can provide enough ENOB, and all the calculations are performed with floating point number. And on the level of hardware implementation, we must consider using fixed point number where ENOB makes important impact on the performance. In this paper, we only limit ourselves in performance evaluation on the level of software, without considering the ENOB impact. However, we added a paragrapgh in the conclusion that “In addition, in a real system, we must consider the hardware implementation efficiency in which we should pay attention to the effective number of bit (ENOB) of analog digital converter (ADC) or digital analog converter (DAC). In this paper, our proposed scheme is verified, only in the level of software calculation with floating point numbers, to be capa-ble of enhancing the performance of decoding algorithm by using appropriate polybinary transformation, and the influence of ENOB is not considered. Therefore, we also need to explore the hardware implementation in future.

4. Can the authors provide a definition of linewidth of laser? Is it Lorentzian linewidth? Or other definition?

Thank you for your comment. We added the definition of linewidth as the full width at half maximum (FHWM) with a sentence “In this paper, the linewidth of laser Δf is defined as the full width at half maximum (FHWM).” on Page 9.

5. In Fig7, there are four sub plots. However, only three plots are described in the figure caption below. I believe that time packing factor of 7(a) is 1, 7(b) is 0.95, 7(c) is 0.65 and 7(d) is 0.45. Please correct the figure caption.

Original:

Figure 7. The normalized frequent and continuous versus real-value of constellation point for original signal, after duobinary transformation signal and after tribinary transformation signal: (a) when the time-packing factor equals 0.95; (b) when the time-packing factor equals 0.65; (c) when the time-packing factor equals 0.45.

Modified:

Figure 7. The normalized frequent and continuous versus real-value of constellation point for original signal, after duobinary transformation signal and after tribinary transformation signal: (a) when the time-packing factor equals 1.00; (b) when the time-packing factor equals 0.95; (c) when the time-packing factor equals 0.60; (d) when the time-packing factor equals 0.45.

6. Are the constellation points in Eq. (3) obtained from the simulation result, like that shown in Fig. 4(c)? Please clarify how one obtain the constellation points in the paper.

Thank you for your comment. The constellation points of Fig.4(c) obtained by simulation. We have added the instructions in our paper.

Original:

Figure 4a shows the PDF diagram of the original QPSK signal when the time-packing factor equals 0.50. In this case, the SE doubles but the ISI is severely induced, with the result that we cannot clearly distinguish the constellation points. Therefore, we utilize the n-level polybinary transformation for the original QPSK signal with the time-packing factor of 0.50. Figure 4b and c show their PDF diagrams after duobinary transformation and tribinary transformation, respectively.

Modified:

Figure 4a shows the PDF diagram of the original QPSK signal when the time-packing factor equals 0.50 by simulation. In this case, the SE doubles but the ISI is severely induced, with the result that we cannot clearly distinguish the constellation points. Therefore, we utilize the n-level polybinary transformation for the original QPSK signal with the time-packing factor of 0.50. Figure 4b and c show their PDF diagrams after duobinary transformation and tribinary transformation by simulation, respectively.

7. In the abstract, there is no need to display so many digits as shown below. Rounding the linewidth*symbolrate to two decimal digits like 1.07e-5, 1.79e-5, should be enough.

Thank you for your comment. This problem has been corrected in paper.

Original:

As the result, the modified-BPS algorithm can cope with the phase noise (PN) with the linewidth´symbol rate at 1.0714E-5, 1.7857E-5, 2.8571E-5 and 3.5714E-5 under time-packing factor of 0.55, 0.50 and 0.45, respectively.

Utilizing modified-BPS algorithm, we find that this method can equalize the PN with the linewidth symbol rate at 1.0714E-5, 1.7857E-5, 2.8571E-5 and 3.5714E-5, in which time-packing factor equals 0.55, 0.50 and 0.45, respectively.

The proposed modified-BPS algorithm can effectively cope with the PN with the Δf·Ts at 1.0714E-5, 1.7857E-5, 2.8571E-5 and 3.5714E-5 under time-packing factor of 0.55, 0.50 and 0.45, respectively.

Modified:

As the result, the modified-BPS algorithm can cope with the phase noise (PN) with the linewidth´symbol rate at 1.07E-5, 1.79E-5, 2.86E-5 and 3.57E-5 under time-packing factor of 0.55, 0.50 and 0.45, respectively.

Utilizing modified-BPS algorithm, we find that this method can equalize the PN with the linewidth symbol rate at 1.07E-5, 1.79E-5, 2.861E-5 and 3.57E-5, in which time-packing factor equals 0.55, 0.50 and 0.45, respectively.

The proposed modified-BPS algorithm can effectively cope with the PN with the Δf·Ts at 1.07E-5, 1.79E-5, 2.86E-5 and 3.57E-5 under time-packing factor of 0.55, 0.50 and 0.45, respectively.

8. In the paper, there are many long sentences. In general, readers can get confused by long sentences. One suggestion is to break down a long sentence into a few short sentences.

Thank you for your comment. We have broken some long sentences into a few short sentences to avoid readers get confused in paper.

Original:

Figure 7a ~ Figure 7d show above mentioned constellation projection PDF graphs cor-responding to the time-packing factors of 1.00, 0.95, 0.65 and 0.45, in which the re-ceived signal are exercised with 1-level polybinary transformation (equivalent to original signal), duobinary transformation and tribinary transformation, respectively.

The original signal has a 4 QAM constellation projection PDF curve, the duobinary transformed signal has a 9 QAM constellation projection PDF curve, and the tribinary transformed signal has a 16 QAM constellation projection PDF curve, as shown in Figure 7a, as the standard benchmark.

When the time-packing factor equals 0.95, which has not much distortion induced by ISI, we observe that all the curves in Figure 7a and b are similar, which means that with the time-packing factor 0.95 we get almost the same performances of the combinations of 1-level polybinary transformation+ MLSE (or MLSE), duobinary transformation + MLSE and tribinary transformation + MLSE, indicating they can all eliminate ISI in FTN system with 0.95 time-packing factor.

However, we see that the duobinary transformed signal curve has the same locations of all peaks, as the case of factor 1.00, as shown in Figure 7a and c, on which we also guess it is appropriate to adopt the combination algorithm of duobinary transformation + MLSE, in order to get a good decoding performance.

We see that the number of curve peaks have changed for the 1-level polybinary transformed and duobinary transformed signals, compared with the case of factor of 1.00, while the curve corresponding to the tribiary transformed signal also have unchanged four peaks, although the curve is apparently deformed.

Modified:

Figure 7a ~ Figure 7d show above mentioned constellation projection PDF graphs cor-responding to the time-packing factors of 1.00, 0.95, 0.65 and 0.45, respectively. Note that, in Figure 7a ~ Figure 7d, the received signal are both exercised with 1-level polybinary transformation (equivalent to original signal), duobinary transformation and tribinary transformation.

The original signal has a 4 QAM constellation projection PDF curve, the duobinary transformed signal has a 9 QAM constellation projection PDF curve, and the tribinary transformed signal has a 16 QAM constellation projection PDF curve, as shown in Figure 7a. In Figure 7a, the three curves as the standard benchmark.

When the time-packing factor equals 0.95, which has not much distortion induced by ISI, we observe that all the curves in Figure 7a and b are similar. It means that with the time-packing factor 0.95 we get almost the same performances of the combinations of 1-level polybinary transformation+ MLSE (or MLSE), duobinary transformation + MLSE and tribinary transformation + MLSE, indicating they can all eliminate ISI in FTN system with 0.95 time-packing factor.

However, we see that the duobinary transformed signal curve has the same locations of all peaks, as the case of factor 1.00, as shown in Figure 7a and c. Hence, we guess it is appropriate to adopt the combination algorithm of duobinary transformation + MLSE, in order to get a good decoding performance.

We see that the number of curve peaks have changed for the 1-level polybinary transformed and duobinary transformed signals, compared with the case of factor of 1.00. And the curve corresponding to the tribinary transformed signal also have unchanged four peaks, although the curve is apparently deformed.

Reviewer 2 Report

1- In line 160 : Even though the assumption of the authors is accepted for the sake of the study, it is not always the case and authors are encouraged to elaborate in general on the effect or interaction of any other impairement with the studied factors.

2- In figure 7 caption : d is missing and I think (a) is with time-packing factor of 1 ...etc.

3- Paragraph beginning in line 329: authors are just stating the number shown in the figure, so I suggest that they rephrase the paragraph describing the results rather than stating numbers.

4- In the conclusion : authors are encouraged to state the next step to develop such technique.

5- There are minor errors:

- (an n-level) all over the manuscript.

- line 255: exact know of ...

Author Response

1. In line 160: Even though the assumption of the authors is accepted for the sake of the study, it is not always the case and authors are encouraged to elaborate in general on the effect or interaction of any other impairement with the studied factors.

Thank you for your comment. We have done some work to equalize extreme polarization impairments in FTN system with over small time-packing factor [1]. In this paper, we propose one scheme to equalize extreme polarization impairments in FTN system with over small time-packing factor. For other impairments such as chromatic dispersion (CD) and frequency offset (FO), we have done a work of equalization envolving all the impairments mentioned above. ([17] P. Sun, X. Zhang, L. Xi, W. Zhang, and X. Tang, "The Effectively Corrected Scheme for Polarization De-multiplexing in Tight Time-Packing PDM-FTN Optical Communication System with Extreme Polarization Impairments," in Asia Communications and Photonics Conference/International Conference, OSA Technical Digest (Optica Publishing Group, 2020), paper M4A.300.) We added this reference, and added a paragraph “We have done a work to jointly equalize PMD and RSOP in FTN system under over small Time-Packing factor [17]. In this paper we only focus our attention to cope with the prob-lem of ISI induced in FTN system. in Page 6.

2. In figure 7 caption: d is missing and I think (a) is with time-packing factor of 1 ...etc.

Thank you for your comment. This problem has been corrected in paper.

Original:

Figure 7. The normalized frequent and continuous versus real-value of constellation point for original signal, after duobinary transformation signal and after tribinary transformation signal: (a) when the time-packing factor equals 0.95; (b) when the time-packing factor equals 0.65; (c) when the time-packing factor equals 0.45.

Modified:

Figure 7. The normalized frequent and continuous versus real-value of constellation point for original signal, after duobinary transformation signal and after tribinary transformation signal: (a) when the time-packing factor equals 1.00; (b) when the time-packing factor equals 0.95; (c) when the time-packing factor equals 0.60; (d) when the time-packing factor equals 0.45.

3. Paragraph beginning in line 329: authors are just stating the number shown in the figure, so I suggest that they rephrase the paragraph describing the results rather than stating numbers.

Thank you for your comment. We have added the analysis in our paper.

Original:

    We can see in Figure 8a ~ Figure.8c that the modified-BPS algorithm can effectively compensate 300 kHz, 500 kHz, 800 kHz and 1000 kHz linewidth with the time-packing factor of 0.45 ~ 0.55. Figure.9a shows that: â‘  When the time-packing factor equals 0.55, the required OSNRs are 15.10 dB, 15.40 dB, 15.50 dB, 15.70 dB and 16.05 dB for the Δf·Ts of 0, 1.07E-5, 1.79E-5, 2.86E-5 and 3.57E-5, respectively; â‘¡ When the time-packing factor equals 0.50, the required OSNRs are 16.10 dB, 16.40 dB, 16.70 dB, 16.90 dB and 17.30 dB for the Δf·Ts of 0, 1.07E-5, 1.79E-5, 2.86E-5 and 3.57E-5, respectively; â‘¢ When the time-packing factor equals 0.45, the required OSNRs are 18.40 dB, 19.10 dB, 19.35 dB, 20.40 dB and 20.90 dB for the Δf·Ts of 0, 1.07E-5, 1.79E-5, 2.86E-5 and 3.57E-5, respectively. The smaller the time-packing factor is, the more ISI induced, and the larger OSNR is required.

    To measure the OSNR penalty, we set the 15.10 dB as the standard benchmark that corresponding to the case of time-packing factor of 0.55 and Δf·Ts of 0. We can see in Figure.9b that: â‘  When the time-packing factor equals 0.55, the OSNR penalties are 0.30 dB, 0.45 dB, 0.60 dB and 0.95 dB for the Δf·Ts of 1.07E-5, 1.79E-5, 2.86E-5 and 3.57E-5, respectively; â‘¡ When the time-packing factor equals 0.50, the OSNR penalties are 1.00 dB, 1.30 dB, 1.60 dB, 1.80 dB and 2.20 dB for the Δf·Ts of 0, 1.07E-5, 1.79E-5, 2.86E-5 and 3.57E-5, respectively; â‘¢ When the time-packing factor equals 0.45, the OSNR penalties are 3.30 dB, 4.00 dB, 4.25 dB, 5.30 dB and 5.80 dB for the Δf·Ts of 0, 1.07E-5, 1.79E-5, 2.86E-5 and 3.57E-5, respectively.

    Hence, we can see that the modified-BPS algorithm can equalize effectively PN in FTN system with over smaller time-packing factor. Meanwhile, using the method of the combination of appropriate n-level polybinary transformation, modified-BPS and MLSE algorithm can effectively recover the received FTN signals.

Modified:

    We can see in Figure 8a ~ Figure 8c that the modified-BPS algorithm can effectively compensate 300 kHz, 500 kHz, 800 kHz and 1000 kHz linewidth with the time-packing factor of 0.45 ~ 0.55. Figure 9a shows that the required OSNRs for the modified-BPS algorithm with the Δf·Ts parameter of 0, 1.07E-5, 1.79E-5, 2.86E-5 and 3.57E-5 when the time-packing factor takes the values in 0.45 ~ 0.55. To observe and compare conveniently, the required OSNRs for modified-BPS algorithm with different time-packing factor and Δf·Ts are listed in Table 1. To measure the OSNR penalty, we set the 15.10 dB as the standard benchmark that corresponding to the case of time-packing factor of 0.55 and Δf·Ts of 0. And Figure 9b shows the OSNR penalties for the modified-BPS algorithm under the same conditions like Figure 9a. And the results are listed in Table 2.

    In Table 1, we can see that required OSNRs are 20.90 dB, 17.30 dB and 16.05 dB for Δf·Ts = 3.57E-5 under the time-packing factor equals 0.45, 0.50 and 0.55, respectively. And corresponding OSNR penalties reach 5.80 dB, 2.20 dB and 0.95 dB which are shown in Table 2, respectively. However, OSNR differences for 0 and 3.57E-5 Δf·Ts are 2.40 dB, 1.20 dB and 0.95 dB under the 0.45, 0.50 and 0.55 time-packing factor, respectively. Therefore, the performances of the modified-BPS algorithm are mainly influenced by the value of time-packing factor rather than Δf·Ts.

    Hence, we can see that the modified-BPS algorithm can equalize effectively PN in FTN system with over smaller time-packing factor. Meanwhile, using the method of the combination of appropriate n-level polybinary transformation, modified-BPS and MLSE algorithm can effectively recover the received FTN signals.

Table 1. The required OSNRs for the modified-BPS algorithm under 0, 1.07E-5, 1.79E-5, 2.86E-5 and 3.57E-5 Δf·Ts with the time-packing factor of 0.45 ~ 0.55.

Table 2. The OSNR penalty for the modified-BPS algorithm under 0, 1.07E-5, 1.79E-5, 2.86E-5 and 3.57E-5 Δf·Ts with the time-packing factor of 0.45 ~ 0.55.

4. In the conclusion: authors are encouraged to state the next step to develop such technique.

Thank you for your comment. We have added the future vision in our paper.

Original:

    In this paper, we revealed the relationship among n-level polybinary transfor-mation, the time-packing factor and the performance of MLSE algorithm. Within a certain range of the time-packing factor, we find that the method of combining appro-priate n-level polybinary transformation and MLSE can effectively eliminate serious ISI in the received FTN signals. With the help of converting 2-dimensional constella-tion diagram into the 1-dimentional constellation projection PDF, we find the reason why we have to choose the different n-level polybinary transformation with the dif-ferent time-packing factor. Coping with the time-packing factor in the large range of 1.00 ~ 0.45, we find the three ranges of it within which the best n-level polybinary we should be adopted. Based on above analysis, we propose a modified-BPS algorithm to compensate PN in FTN system with the extreme small time-packing factor. The pro-posed modified-BPS algorithm can effectively cope with the PN with the Δf·Ts at 1.0714E-5, 1.7857E-5, 2.8571E-5 and 3.5714E-5 under time-packing factor of 0.55, 0.50 and 0.45, respectively. In the case of above conditions, the SE can be enhanced up to about 3.27 bit/s/Hz, 4 bit/s/Hz and 4.88 bit/s/Hz, respectively.

Modified:

    In this paper, we revealed the relationship among n-level polybinary transformation, the time-packing factor and the performance of MLSE algorithm. Within a certain range of the time-packing factor, we find that the method of combining appropriate n-level polybinary transformation and MLSE can effectively eliminate serious ISI in the received FTN signals. With the help of converting 2-dimensional constellation diagram into the 1-dimentional constellation projection PDF, we find the reason why we have to choose the different n-level polybinary transformation with the different time-packing factor. Coping with the time-packing factor in the large range of 1.00 ~ 0.45, we find the three ranges of it within which the best n-level polybinary we should be adopted. Based on above analysis, we propose a modified-BPS algorithm to compensate PN in FTN system with the extreme small time-packing factor. The proposed modified-BPS algorithm can effectively cope with the PN with the Δf·Ts at 1.07E-5, 1.79E-5, 2.86E-5 and 3.57E-5 under time-packing factor of 0.45, 0.50 and 0.55, respectively. In the case of above conditions, the SE can be enhanced up to about 4.88 bit/s/Hz, 4 bit/s/Hz and 3.27 bit/s/Hz, respectively. It is noteworthy that, to verify the performance of our proposed scheme in FTN system with the extreme small time-packing factor, we assume FO, CD, PMD, RSOP and other impairments are not exist or have been equalized. However, in a large accumulated CD system such as long-haul transoceanic optical fiber communication system, we will face the trouble problem of enhanced phase noise (EEPN) which is caused by the conflict between the long CD response time and short time along with the laser phase noise variation. Unlike CD or PMD impairments, there is not an effective way to mitigate it [18]. We should pay attention to the EEPN trouble in our next works. In addition, in a real system, we must consider the hardware implementation efficiency in which we should pay attention to the effective number of bit (ENOB) of analog digital converter (ADC) or digital analog converter (DAC). In this paper, our proposed scheme is verified, only in the level of software calculation with floating point numbers, to be capable of enhancing the performance of decoding algorithm by using appropriate polybinary transformation, and the influence of ENOB is not considered. Therefore, we also need to explore the hardware implementation in future. ([18] Han. Sun, Mehdi Torbatian, Mehdi Karimi, et al., "800G DSP ASIC Design Using Probabilistic Shaping and Digital Sub-Carrier Multiplexing," J. of Lightwave Technol. 38, 4744-4756 (2020).)

5. There are minor errors:

- (an n-level) all over the manuscript.

- line 255: exact know of ...

Thank you for your comment. This problem has been corrected in paper.

Original:

Based on the above analysis, we find that the method of making a n-level polybinary transformation combined with a designed MLSE has the better performance than that without doing so.

A n-level polybinary transformation can be completed by a series of delay operations and modulo-two sum operations.

A n-level polybinary transformation has its name in term of aforementioned n-2 symbol period delays, in which the 2-level and 3-level polybinary transformation have their conventional names of duobinary transformation and tribinary transformation, respectively.

Generally, the most effective algorithm is the BPS algorithm which can equalize PN for QPSK signal and other QAM signal, such as 16QAM, 32QAM, etc. BPS can achieve optimal performance based on the exact know the constellation point’s locations of the QAM signals.

Modified:

Based on the above analysis, we find that the method of making an n-level polybinary transformation combined with a designed MLSE has the better performance than that without doing so.

An n-level polybinary transformation can be completed by a series of delay operations and modulo-two sum operations.

An n-level polybinary transformation has its name in term of aforementioned n-2 symbol period delays, in which 2-level and 3-level polybinary transformation have their conventional names of duobinary transformation and tribinary transformation, respectively.

Generally, the most effective algorithm is the BPS algorithm which can equalize PN for QPSK signal and other QAM signal, such as 16QAM, 32QAM, etc. BPS can achieve op-timal performance based on the exact know of the constellation point’s locations of the QAM signals.

Round 2

Reviewer 1 Report

The revised manuscript addressed my comments and suggestions.  The authors also corrected a few errors presented in the first draft.  In my opinion, the paper can be published in the current form.  

The results are obtained from theoretical simulation.  It is all right to publish the initial result with simulation data.  However, the applicability to real-world scenarios cannot be validated without experimental results.  Further work should be complemented with experimental results.